# Machine Learning Models for Predicting Water Quality of Treated Fruit and Vegetable Wastewater



**Gurvinder Mundi [1], Richard G. Zytner [2,*], Keith Warriner [3], Hossein Bonakdari [4] and Bahram Gharabaghi [2]**

1    Mueller (Echologics Division), Toronto, ON M9W 1B3, Canada; gmundi@uoguelph.ca
2    School of Engineering, University of Guelph, Guelph, ON N1G 2W1, Canada; bgharaba@uoguelph.ca
3    Department of Food Science, University of Guelph, Guelph, ON N1G 2W1, Canada; kwarrine@gmail.com
4    Department of Soils and Agri-Food Engineering, Laval University, Québec, QC G1V 0A6, Canada; hossein.bonakdari@fsaa.ulaval.ca
*    Correspondence: rzytner@uoguelph.ca; Tel.: +519-824-4120 (ext. 53859)

**Abstract:** Wash-waters and wastewaters from the fruit and vegetable processing industry are characterized in terms of solids and organic content that requires treatment to meet regulatory standards for purpose-of-use. In the following, the efficacy of 13 different water remediation methods (coagulation, filtration, bioreactors, and ultraviolet-based methods) to treat fourteen types of wastewater derived from fruit and vegetable processing (fruit, root vegetables, leafy greens) were examined. Each treatment was assessed in terms of reducing suspended solids, total phosphorus, nitrogen, biochemical and chemical oxygen demand. From the data generated, it was possible to develop predictive modeling for each of the water treatments tested. Models to predict post-treatment water quality were studied and developed using multiple linear regression (coefficient of determination ($R^2$) of 30 to 83%), which were improved by the generalized structure of group method of data handling models ($R^2$ of 73–99%). The selection of multiple linear regression and the generalized structure of group method of data handling models was due to the ability of the models to produce robust equations for ease of use and practicality. The large variability and complex nature of wastewater quality parameters were challenging to represent in linear models; however, they were better suited for group method of data handling technique as shown in the study. The model provides an important tool to end users in selecting the appropriate treatment based on the original wastewater characteristics and required standards for the treated water.

**Keywords:** water quality; multiple linear regression (MLR); wastewater; machine learning; wash-water; food processing; treatment feasibility; water resources

## 1. Introduction

Fruit and vegetable food processing is a water-demanding process with the main uses being in washing, cutting, peeling, sanitizing, cooling, transporting, and equipment and machinery cleaning [1]. Different quality waters are used in different stages; however, the highest quality waters, that of the potable standard, is required to be used for final cleaning/rinsing. The high water usage ultimately leads to the generation of large volumes of wastewater [2,3]. In many cases, the wastewater (used once and wasted) or wash-water (recirculated waters reused for washing) are often high in solids, biochemical oxygen demand (BOD), and other pollutants [4,5]. The generated wastewater and wash-waters require adequate treatment and disinfection before they are either reused in the process or disposed of to reduce environmental impact.

Adequate treatment of waters is necessary to provide proper food safety when water is reused within the process and to protect the environment from receiving excess loads of contamination [6]. The recirculation and reuse of the wash-water are important in reducing the amount of wastewater requiring treatment or disposal in an overall program.

Thus, water reclamation and reuse are often practiced to minimize the amount of water utilized in many water-intensive industries like the fruit and vegetable washing and processing sector.

There are a diverse range of wastewater treatments available that have variable efficacy in terms of removing soluble and insoluble constituents. Moreover, the degree to which wastewater needs to be treated depends on the end-use, for example, disposal or re-introduction into the process. Selecting which treatment to apply is challenging due to the lack of comprehensive datasets and models for individual or combination of methods for different wastewater types. Information and models/tools that address treatment, water reuse feasibility, and its prediction are lacking.

Multiple linear regression (MLR) methods are popularly used for predicting post-treatment water quality parameters [7,8]. Due to the multiplicity of factors affecting post-treatment water quality, these linear data analysis methods may not be robust enough to deal with the complex data patterns in such problems. Several machine learning techniques can be a promising approach to cope with the limitations of MLR methods. Due to simple architecture, artificial neural networks (ANNs) are a good mathematical model alternative that have been used widely in recent studies to solve nonlinear problems in the science and engineering sector [9–11]. Granata et al. [12] highlighted that machine learning algorithms and effective use of field data are useful in modelling water quality parameters, which then can be considered for the sizing of the treatment unit [13]. Furthermore, different optimization methods were developed to improve the quality and sustainability in wastewater treatment process [14–16]. For the problems with high correlated input parameters, this technique is very effective and easy to implement. However, they are not able to generate an explicit equation between input parameters and output variable(s) that are much needed in engineering problems for practical application [10,17,18].

Among the various intelligence methods, a reliable and self-organizing neural network sub-model is the group method of data handling (GMDH) method. This method is successfully applied in diverse nonlinear problems such as plant disease detection [19], pressure meter modulus [20], soil temperature [21], dispersion coefficients in water pipelines [22], and small strain shear modulus of grouted sands [23]. Although the modeling competence of ANN-based techniques has been well documented, very few GMDH-based models with application in post-treatment water quality parameters analysis have been reported in recent literature. Thus, the objective of this study is to address the knowledge gap within the industry by developing models using well-known MLR techniques and the more recent GMDH-based methods. Producers and government are all interested in knowing the efficiency of various treatment processes, but information is lacking in the agriculture sector.

The proposed models highlighted in this study will predict post-treatment (treated) water quality parameters for total suspended solids ($TSS_{Treated}$), chemical oxygen demand ($COD_{Treated}$), biological oxygen demand ($BOD_{Treated}$), total nitrogen ($TN_{Treated}$), total phosphorus ($TP_{Treated}$), and Ammonia as Nitrogen ($NH4\text{-}N_{Treated}$), based on operation type, raw water quality, and wastewater treatment process (such as settling, dissolved air flotation, and membrane bioreactors).

The predicted post-treatment water quality parameters can be utilized to address treatment options and to assess the potential for meeting regulatory compliance for wastewater disposal and reuse, and to better understand the impact of releasing treated waters into the environment and on sensitive organisms downstream. The models will be of great use to all stakeholders, such as farmers, producers, processors, technology providers, consultants, and regulators to detect the level of contamination in wastewater before disposal or reuse.

## 2. Materials and Methods

### 2.1. Study Area, Wash-Water Samples, and Laboratory Analysis

The study was carried out in Southern Ontario, Canada. The selected wastewaters and wash-waters were collected from various fruit and vegetable washing and processing

facilities, including fresh-cut produce (apple, çarrot, potato, Ginseng, and others). Many of the facilities were on-farm operations, currently utilizing passive treatments such as settling ponds, while other samples were collected from urban facilities, which dispose of wastewaters in the municipal Sewer. Wash-waters were generated in many applications where fruits and vegetables were washed to remove soil, processed, and/or provide microbiological decontamination. Different processing steps were required based on the type of vegetable or fruit being processed.

Root vegetables such as carrots, potatoes, and ginseng require washing to clean and remove soils that are attached to root material. This process is stated as washing (W). However, whenever the root vegetable or another fruit or vegetable requires processing, such as cutting/peeling, then the process is stated as washing and processing (WP). In addition, the products were also classified as a root vegetable, tree fruit, leafy green, and above ground. The full dataset consists of four independent subsets, two of which were studied and presented in Mundi et al. [24], while the remaining two were introduced for characterization in Mundi et al. [25], which were collected by the Ontario Ministry of Agriculture Finance and Rural Affairs (OMAFRA). The results presented in Mundi et al. [24] were table-based or decision matrices, where the user would read the treatment combination of the charts. These decision matrices will now be converted to models. Additional data from OMAFRA, in combination with Mundi et al. [25], were used to develop Power-Rank models for characterizing water quality to fill data gaps and develop models as presented herein.

A total of 245 unique samples were contained in the master dataset, 223 contained data on bench scale treatments, while the other 22 related to full-scale treatments. The bench-scale treatments consisted of screening (S), hydrocyclone (HC), the settling using jar test method with an optimized coagulation and flocculation process (C&F), dissolved air flotation using optimized coagulant dosage (DAF), centrifuge (C), and electrocoagulation (EC&F). Full-scale treatments sampled included single tank settling (SET1), settling followed by open grass (SET1G), three settling tanks in series (SET3), pond (POND), sequential batch reactor with four stages—settling, aeration, nutrient removal, settling (SETBIO4)— and membrane bioreactor with reverse osmosis and UV disinfection (MBR+RO+UV). The samples were collected at random from each facility under normal operating conditions. The samples collected consisted of (1) raw wash-water, the wastewater that is produced by the washing and/or processing operations, and (2) post-onsite treatment, the effluent treated wastewater employed by the facility. The raw wash-water and the post-onsite treatment samples were analyzed for a suite of water quality tests.

In addition, the raw wash-waters were also treated with six bench-scale treatments. These include the following treatments: Settling (jar test control—1 min rapid mix at 100 rpm, 10 min slow mix at 30 rpm, and 20 min settling), settling with a coagulation and flocculation process (jar test with coagulants varying in dosage from 5 to 400 mg/L, 1 min rapid mix at 100 rpm, 10 min slow mix at 30 rpm, and 20 min settling), screening (sieve 100 um), centrifugation (1801 × G for 3-min), dissolved air flotation (10-min retention of recycling water at a rate of 50% and 10-min detention for flotation) using an optimized coagulant dosage as determined earlier using the jar test, hydro-cyclone (4 mm apex, 6.7 mm vortex finder, 48 mm diameter, and 1.3 L/min of flow), and electrocoagulation (Maximum of 575.1 $cm^2$ surface area, minimum reaction time of 10-min, and maximum of 0.27 $kWh/m^3$ power consumption). A detailed description of bench-scale treatments and their methodology, in addition to full-scale treatments, are outlined in Mundi et al. [24].

Water quality parameters were tested using standard methods as listed in text Standard Methods for Examination of Water and Wastewater, 22nd Edition [26]. TS and TSS were measured using the evaporation–mass balance Method 2540 B and Method 2540 D, respectively. BOD was measured using the Standard Method 5210. Additional water quality testing was conducted using the Hach instrumentation and water quality testing kits, such as COD (Dichromate Method, TNT821,2—Method 8000), TN (Persulfate Digestion, TNT826,7,8—Method 10208), TP (Ascorbic Acid, TNT843,4,5—Method 10209 and

10210), and Ammonia as N (Salicylate, TNT830,1,2—Method 10205). The Digital Reactor Block from HACHCO (DRB200-02) and the Ultraviolet-Visible Spectrophotometer from HACHCO (DR5000-03) were used to complete the previously listed tests.

### 2.2. Multiple Linear Regression

The compiled master dataset was manually inspected, and missing data were predicted using the relationships developed in Mundi et al. [25], between similar water quality parameters. Dataset statistics and modelling of MLR was completed using R software (RStudio—Version 1.0.153). A variable selection technique can reduce the number of input variables needed for developing models. This is done to remove input variables that have no significant relationship with the output variable, which reduces the computational complexity and improves predictions [8]. Using too many input variables can lead to an overfitted model and makes it less practical, as the collection and analysis of additional variables can be costly. Variable selection in MLR models was achieved through Pearsons' Correlation matrix, highlighting a statistically significant ($p$-value < 0.01) correlation.

Before modeling the data, the inputs and output were normalized using a linear scaling method in Equation (1), with a 0 to 1 scale [27]. Equation (1) was used for numerical variables, such as BOD; however, the normalizing of categorical variables such as process and treatment were different. The process and treatment were scaled using a ranking number, similar to Mundi et al. [25], obtained by the ranking of average treatment reduction efficiency with respect to treatment parameters, such as BOD [25]. Normalizing was done to prevent the magnitude of each parameter from potentially influencing the weights assigned in model development, as the dataset includes several different types of measurements.

$$z_i = \frac{x_i}{x_{max} + 1} \tag{1}$$

where $z_i$ is the normalized value computed for input $i$, $x_i$ is the actual value of the input $i$, and $x_{max}$ is the maximum value of all input values for a given parameter. MLR is a statistical technique used to model the relationship between two or more explanatory variables (independent) and a response variable (dependent) by fitting a linear equation to the observed data. The MLR model can be defined as:

$$y_i = \beta_0 + \beta_1 x_{i,1} + \beta_2 x_{i,2} + \ldots + \beta_k x_{i,k} + \varepsilon_i \tag{2}$$

where $y_i$ is the dependent variable, $\beta_0$ is a constant or intercept, $x_{i,k}$ is an independent variable, $\beta_k$ is the coefficient regression vector or slope, and $\varepsilon_i$ is random measured error. In the present study, R language/software (RStudio—Version 1.0.153) was used to calculate the MRL models.

### 2.3. Generalized Structure of Group Method of Data Handling (GSGMDH)

The water quality of fruit and vegetable wastewater treatment systems is a complex multivariable system that is not easily modelled with theoretical or analytical variable-based models. The group method of data handling (GMDH) can be employed effectively in real-world cases where there is no theoretical experience about the association between the input variables and the outcome [18,19]. The actual variable ($y$) for the input vector ($x_1$, $x_2$, $x_3$, ..., $x_n$) with a given M observation ($i = 1, 2, \ldots$, M) can be defined as follows:

$$y_i = f(x_{i,1}, \ x_{i,2}, \ x_{i,3}, \ \ldots, \ x_{i,n}) \tag{3}$$

GMDH can be trained to estimate the outcome value ($\hat{y}_i$) for given input variables as follows:

$$\hat{y}_i = \hat{f}(x_{i,1}, \ x_{i,2}, \ x_{i,3}, \ \ldots, \ x_{i,n}) \tag{4}$$

The objective function is to minimize the square difference between the actual outcome ($y$) and estimated $\hat{y}_i$, as follows:

$$Min\left\{\sum_{i=1}^{m}\left(\hat{f}(x_{i,1},\ x_{i,2},\ x_{i,3},\ \dots,\ x_{i,n})-f(x_{i,1},\ x_{i,2},\ x_{i,3},\ \dots,\ x_{i,n})\right)^2\right\} \tag{5}$$

The basis of the GMDH algorithm is the process of constructing a high-order polynomial known as the Volterra functional series, as follows:

$$y=a_o+\sum_{i=1}^{m}a_ix_i+\sum_{i=1}^{m}\sum_{j=1}^{m}a_{ij}x_ix_j+\sum_{i=1}^{m}\sum_{j=1}^{m}\sum_{k=1}^{m}a_{ijk}x_ix_jx_k+\dots \tag{6}$$

Hence, in the GMDH algorithm, the series of Volterra functions are decomposed into quadratic binomial polynomials. This mathematical description can be simplified by quadratic polynomials consisting of only two variables (neurons) in the form of:

$$\hat{y}_i=\hat{f}\left(x_i,x_j\right)=a_o+a_1x_2+a_2x_j+a_3x_i^2+a_4x_j^2+a_5x_ix_j \tag{7}$$

In the GMDH algorithm, a regression polynomial is yielded according to Equation (7) for all possibilities consisting of two independent variables to create the best fit between the M observed values by applying objective function as presented in Equation (5).

Although convectional GMDH has a high capability for nonlinear problem modelling the following points have an important impact on obtained results:

1.  The second-order polynomial defined polynomial structure (Equation (7)) has only two input neurons.
2.  The input neurons in each layer are selected only from adjacent layers.

According to these two limitations, problems of high complexity, the use of second-order polynomials may not yield acceptable outcomes in GMDH. Furthermore, considering two inputs for each neuron leads to an increase in the number of neurons to reach an adequate model. The use of adjacent neuron layers increases the number of polynomials produced. Therefore, these issues have a significant impact on the accuracy and simplicity of the proposed models. Thus, in this study, a generalized structure of GMDH (GSGMDH) is presented [17,18]. We coded the GSGMDH mode in the MATLAB software environment. The proposed model modifies the general convectional structure of the GMDH so that it simultaneously investigates all possible modes of achieving the best and simplest model available by using polynomials of order 2 and 3 as well as the use of two and three neurons. Finally, it selects the best model using the input index for each corrected Akaike information criterion (*AIC*) as follows:

$$AIC=n\times log\left[\sum_{i=1}^{n}(\hat{y}_i-y_i)^2+2D+\frac{2D(2D+1)}{N-D-1}\right] \tag{8}$$

where, $n$ is the number of samples, $\hat{y}_i$ and $y_i$ are predicted and observed values, and $D$ is the number of tuned variables through modeling with GMDH.

Four situations can occur: (1) Second-order polynomials, (2) second-order polynomials with three inputs, (3) third-order polynomials with two inputs, and (4) third-order polynomials with three inputs. Among these four states, the first state is precisely the same relationship provided for the convectional GMDH (Equation (5)). Therefore, the general form of the polynomial defined in this study is as follows:

$$\begin{aligned}\hat{y}=a_o+a_1x_k\ &+a_2x_q+a_3x_p+a_4x_qx_k+a_5x_px_k+a_6x_px_q+a_7x_k^2+a_8x_q^2\\&+a_9x_p^2+a_{10}x_px_qx_k+a_{11}x_qx_k^2+a_{12}x_k^2x_k+a_{13}x_px_k^2\\&+a_{14}x_px_q^2+a_{15}x_p^2x_k+a_{16}x_p^2x_q+a_{17}x_k^3+a_{18}x_q^3+a_{19}x_p^3\end{aligned} \tag{9}$$

where $k,p,q\in\{1,2,3,\dots,n\}$.

The flowchart of the GSGMDH method is presented in Figure 1. As shown in this figure, after definition of the initial value for GSGMDH model parameters, all possible neurons will be created. In this operation, in order to select the final model, two criteria were checked. The accuracy of the obtained results was verified (1) with *RMSE* to keep the maximum allowable neurons and remove other neurons and (2) with *AIC* to examine the simplicity of the designed architecture.

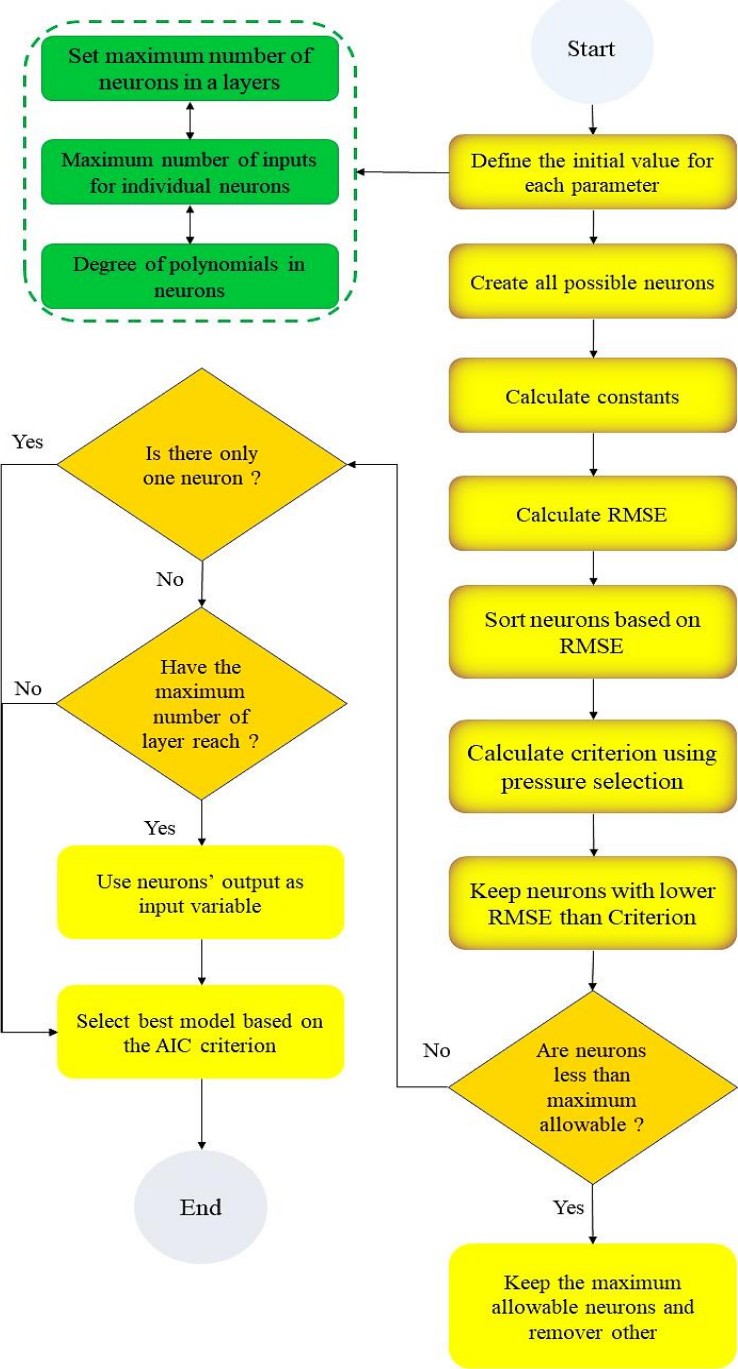

**Figure 1.** The flowchart of the proposed GSGMDH.

*2.4. Model Performance Evaluation*

A variety of statistical measures were utilized to understand the unique properties of the model performance. The coefficient of correlation (*R*) was used to understand the

amount of observed variance within the models, shown in Equation (10). The root mean square error (*RMSE*) and mean absolute percent error (*MAPE*) was used to understand model accuracy and precision, as shown in Equations (11) and (12). *RMSE* shows differences between the observed and predicted values in the units of the variable of the study. In Equations (10) and (11), variables *O* and *P* are the observed and model predicted values, respectively, and *n* is the number of observations.

$$R = \frac{\sum_{i=1}^{n}(O_i - \overline{O})(P_i - \overline{P})}{\sqrt{\sum_{i=1}^{n}(O_i - \overline{O})^2 * \sum_{i=1}^{n}(P_i - \overline{P})^2}} * 100 \tag{10}$$

$$RMSE = \sqrt{\frac{\sum_{i=1}^{n}(O_i - P_i)^2}{n}} \tag{11}$$

$$MAPE = \frac{100}{n}\sum_{i=1}^{n}\left|\frac{(O_i - P_i)}{O_i}\right| \tag{12}$$

### 3. Results and Discussion

In this study, for the first time, the results of wastewater treatment feasibility, from bench-scale testing and full-scale treatment sampling of fruit and vegetable washing and processing facilities were analyzed to develop predictive models to estimate water quality parameters of the treated wastewater. The study also used facility operating variables as input variables, such as the type of process (washing versus processing). The optimal subsets of input variables for the different models were selected using correlation analysis and statistical significance tests. The MRL and GSGMDH modelling techniques did not consider any deterministic models, chemical, or biological knowledge about the treatment processes, and were based on mathematical grounds only. MLR and GSGMDH have been used before; however, this study used these modelling techniques for wastewater and wash-waters treatment prediction for a wide range of fruits and vegetables. Previous studies by Chen et al. [28], Gil et al. [2], Kern et al. [5], and Mjalli et al. [29] have only focused on individual/single fruit or vegetable wash-water or wastewaters. This study provides insight into three novel aspects, a new sector with respect to industrial/agricultural wastewaters, the use of bench-scale testing and full-scale treatment data, and the large variety of fruit and vegetable wash-waters including the utilization of process type (W versus WP).

The first step was to transform categorical variables to numerical values for treatment and process variables, done using ranking analysis as described in the methodology. The aggregated ranks corresponding to treatment and process type (W or WP) are noted in Table 1. The ranks, best (1) to worst (0), indicate how different treatments impact the removal of the studied water quality parameters. Ranks greater than 0.75 indicate the most effective treatments, such as C&F, EC&F, and DAF for bench scale and MBR+RO+UV, POND, and SET1G for full scale. Ranks lower than 0.25 show the least effective treatments, while the remainder of the range 0.74–0.26 indicates moderate treatment effectiveness. TSS removal was least effective for S and HC treatment compared to C&F and settling at full scale, which utilizes chemicals and long settling time, respectively. Similar conclusions can be drawn for other water quality parameters under different studied treatments from Table 1. Some key observations show that MBR was the best overall; this was no surprise as MBR treatments produce the highest quality waters regardless of the type of wash-water or process. However, MBR is energy-intensive, requires long start-up times, and are very sensitive to changes in wastewater feed. EC&F and C&F show good reduction across many water quality parameters. Full-scale treatments with pond, settling with grasslands, and settling with three tanks in series (POND, SET1G, and SET3) were capable of reducing solids effectively, under the right conditions (flow, concentrations, and settling time).

**Table 1.** The developed ranks represent the ability of treatment to be effective, ranked best (1) to worst (0). Ranks greater than 0.75 are bolded, and ranks lower than 0.25 are shaded, while the remainder are left blank, for visual aid. The ranking for process is also provided.

| | | TSS_Treated | | TP_Treated | | TN_Treated | | COD_Treated | | BOD_Treated | | NH4-N_Treated | |
|---|---|---|---|---|---|---|---|---|---|---|---|---|---|
| | **Process\** | **W** | **WP** | **W** | **WP** | **W** | **WP** | **W** | **WP** | **W** | **WP** | **W** | **WP** |
| | **Treatments** | **1** | **0.5** | **1** | **0.5** | **1** | **0.5** | **1** | **0.5** | **1** | **0.5** | **0.5** | **1** |
| Bench Scale | C | 0.55 | 0.70 | **0.11** | 0.29 | 0.44 | **0.14** | 0.33 | 0.29 | 0.33 | **0.13** | 0.33 | **0.14** |
| | DAF | 0.73 | 0.80 | 0.67 | 0.43 | 0.67 | 0.29 | 0.56 | 0.43 | **0.22** | **0.25** | 0.44 | 0.29 |
| | EC&F | 0.45 | 0.60 | 1 | 0.57 | 0.78 | 0.57 | 0.89 | 0.86 | 0.56 | 0.75 | 0.56 | 0.71 |
| | C&F | 1 | 0.90 | 0.44 | 0.86 | 0.56 | 0.43 | 0.67 | 0.63 | 0.44 | 0.5 | 0.22 | 0.38 |
| | HC | 0.27 | 0.50 | - | - | - | - | - | - | - | - | - | - |
| | S | **0.09** | **0.10** | - | - | - | - | - | - | - | - | - | - |
| Full Scale | MBR + RO + UV | - | 1 | - | 1 | - | 1 | - | 1 | - | 1 | - | 1 |
| | POND | **0.18** | 0.40 | 0.89 | - | 0.89 | - | **0.22** | **0.14** | 0.78 | 0.38 | 0.89 | 0.57 |
| | SET1 | 0.36 | 0.30 | 0.33 | 0.71 | 0.33 | 0.86 | 0.44 | 0.57 | 0.89 | 0.63 | 0.78 | 0.86 |
| | SET1G | 0.91 | - | 0.78 | - | 1 | - | 0.78 | - | 1 | - | 0.67 | - |
| | SET3 | 0.82 | **0.20** | 0.22 | **0.14** | **0.22** | 0.71 | 1 | 0.71 | 0.67 | **0.88** | 1 | 0.43 |
| | SETBIO4 | 0.64 | - | 0.56 | - | **0.11** | - | **0.11** | - | **0.11** | - | **0.11** | - |

Washing (W), Washing and processing (WP) Bench-scale treatments consisted of Screening (S), hydrocyclone (HC), Settling using Jar Test method with optimized coagulation and flocculation process (C&F), dissolved air flotation using optimized coagulant dosage (DAF), centrifuge (C), and electrocoagulation (EC&F). Full-scale treatments sampled included a single tank settling (SET1), settling followed by open grass (SET1G), three settling tanks in series (SET3), pond (POND), sequential batch reactor with four stages—settling, aeration, nutrient removal, settling (SETBIO4), and mem-brane bioreactor with reverse osmosis and UV disinfection (MBR + RO + UV). (-) test not applicable.

There are some differences between W and WP within the same water quality parameter and treatment, for example, C&F for $TP_{Treated}$ show, good effectiveness for WP but not so great for W type processes. Some process types and water quality parameters were not part of the study, such as the W type wash-waters for MBR. The missing water quality parameters for S and HC treatments were not collected since these treatments are least likely to impact TP, TN, COD, BOD, and $NH_4$-N, as shown in the literature. The ranks in Table 1 require substitution into the MLR equations (Table 2) for estimating treated water quality parameters. More information on the MLR equations is provided later.

**Table 2.** MLR equations for predicting treatment effluent water quality of wash-water treatments.

| MLR Equations | Equation Number |
|---|---|
| $TSS_{Treated} = 0.095 - 0.191(Treatment) + 0.011(Process) + 0.136(TS_{Raw}) + 0.131(TN_{Raw})$ | (13) |
| $BOD_{Treated} = 0.013 - 0.043(Treatment) + 0.025(Process) - 0.031(TDS_{Raw}) + 0.563(BOD_{Raw}) - 0.006(TN_{Raw})$ | (14) |
| $COD_{Treated} = 0.237 - 0.169(Treatment) - 0.172(Process) + 0.270(TDS_{Raw}) + 0.166(COD_{Raw})$ | (15) |
| $TP_{Treated} = 0.035 - 0.069(Treatment) - 0.172(Process) + 0.318(TP_{Raw}) + 0.282(NH_4 - N_{Raw})$ | (16) |
| $TN_{Treated} = 0.1202 - 0.236(Treatment) + 0.059(Process) - 0.311(TDS_{Raw}) - 0.07(TN_{Raw}) + 0.620(NH_4 - N_{Raw})$ | (17) |

After converting all data to numerical values, it was assessed for Pearson's Correlation analysis, highlighting statistically significant ($p$-value < 0.01) correlations, presented in Table 3. Correlation analysis is a valuable tool in identifying correlations between water quality parameters, process, treatments, and dependent variables [29]. More importantly, it shows which input parameters have a good relationship with the dependent variable. Tomperi et al. [7] also used linear correlation analysis to identify variables that impact each other, similarly, BOD and COD have the highest correlation, while other parameters did not show similar results. This is due to the wide variety of wash-waters/wastewater explored in this study compared to the single source, pulp, and paper mill in studies by Tomperi et al. [7]. Treatment was relevant for most water quality parameters; however, for $BOD_{Treated}$ and $COD_{Treated}$, the process was more relevant, suggesting the process has higher control as compared to the treatment, with respect to treated waters. Many core

raw water quality parameters were relevant as expected, which are defined, for example, $BOD_{Raw}$ being the core parameter for the $BOD_{Treated}$ model. The correlations were weak for the $TSS_{Treated}$ model, as the highest correlation was observed to be 0.32 with $TS_{Raw}$ and the lowest was 0.08 with process type, which were similar to the relationships shown in Mjalli et al. [29] and Tomperi et al. [7] between water quality parameters. These correlations were used as the basis for selecting the different input parameters for the studied models. The authors selected variables with significant relationship for use in MLR and GSGMDH as input water quality parameters.

**Table 3.** The correlation matrix between independent and dependent variables.

|  | **Process** | **Treatment** | **$pH_{Raw}$** | **$BOD_{Raw}$** | **$COD_{Raw}$** | **$NH_4$-$N_{Raw}$** | **$TN_{Raw}$** | **$TP_{Raw}$** | **$TSS_{Raw}$** | **$TS_{Raw}$** | **$TDS_{Raw}$** |
|---|---|---|---|---|---|---|---|---|---|---|---|
| $TSS_{Treated}$ | −0.08 | −0.42 | 0.02 | −0.04 | 0.16 | 0.05 | 0.22 | 0.13 | 0.24 | 0.32 | 0.05 |
| $BOD_{Treated}$ | −0.53 | −0.29 | −0.2 | 0.9 | 0.62 | 0.09 | 0.36 | 0.04 | 0.05 | 0.28 | 0.61 |
| $COD_{Treated}$ | −0.58 | −0.26 | −0.11 | 0.65 | 0.56 | 0.07 | 0.19 | −0.01 | 0.04 | 0.40 | 0.62 |
| $TP_{Treated}$ | −0.55 | −0.35 | −0.19 | 0.07 | 0.16 | 0.48 | 0.19 | 0.88 | 0.08 | 0.07 | 0.02 |
| $TN_{Treated}$ | −0.23 | −0.43 | 0.17 | 0.34 | 0.49 | 0.58 | 0.50 | 0.40 | 0.10 | 0.27 | 0.35 |
| $NH_4$-$N_{Treated}$ | −0.13 | −0.46 | −0.07 | 0.18 | 0.41 | 0.85 | 0.37 | 0.27 | 0.56 | 0.14 | −0.05 |

The range for wash-water quality and treated water quality parameters are shown in Table 4, highlighting the minimum, maximum, and the average values of the inputs and outputs used for the different models. The parameters studied and its values are in line with those studied by Letho et al. [1] and Gil et al. [2]; however, some wash-water studied had excessive levels of TSS due to soils from root vegetables. Disposal of wastewater and wash-water requires meeting the listed regulatory standards, also shown in Table 4.

**Table 4.** Model input and output parameters and their minimum, maximum, and mean values.

| Raw Wash-water Quality Parameters | (mg/L) | $TSS_{Raw}$ | $TDS_{Raw}$ | $TS_{Raw}$ | $COD_{Raw}$ | $BOD_{Raw}$ | $TN_{Raw}$ | $TP_{Raw}$ | $NH_4$-$N_{Raw}$ |
|---|---|---|---|---|---|---|---|---|---|
|  | min | 24 | 364 | 468 | 20 | 5 | 0.9 [a] | 1.30 | 0.09 |
|  | mean | 2498 | 1532 | 3795 | 1556 | 387 | 15 | 16.5 | 3.1 |
|  | max | 42,920 | 8740 | 13,855 | 12,400 | 3760 | 101 | 179 | 35 [b] |

| Treated Wash-water Quality Parameters | (mg/L) | $TSS_{Treated}$ | $COD_{Treated}$ | $BOD_{Treated}$ | $TN_{Treated}$ | $TP_{Treated}$ | $NH_4$-$N_{Treated}$ |
|---|---|---|---|---|---|---|---|
|  | min | 0 | 2 | 2 | 0.03 | 0.04 | 0 |
|  | mean | 452 | 632 | 177 | 9.4 | 5.6 | 4.5 |
|  | max | 7160 | 8300 | 2300 | 53.1 | 90 | 70 |

| Effluent requirements for wastewater discharge in Canada | (mg/L) | TSS | TDS | TS | BOD | TP | NH4-N |
|---|---|---|---|---|---|---|---|
|  | Drinking Water [c] | - | 500 | - | - | 0.01 | 0.02 [g] |
|  | Sanitary /Sewer Discharge [d] | 350 | - | - | 300 | 10 | - |
|  | PWQO [e] | 25 [f] | - | - | 20 [f] | 0.02 |  |

[a] For $TN_{Treated}$ model lower limit for TN is 2.5 mg/L, [b] For $NH_4$-$N_{Treated}$ model higher limit for $NH_4$-N is 105 mg/L, [c] Data obtained from Supporting Document for Ontario Drinking Water Quality Standards, Objectives and Guidelines, Tables 1, 3 and 5, [d] Data obtained from City of Toronto Sewer Discharge and Storm Water Discharge Limits, Table 1, [e] Data obtained from Provincial Water Quality Objectives for Surface Water, some parameters are subjected to additional conditions, [f] Limits for effluent discharged to receiving waters; Guidelines for Effluent Quality and Wastewater Treatment at Federal Establishments, [g] Additional requirements related to pH.

**Table 5.** Models generated for treated water quality parameters showing the input variables used and their corresponding $R^2$ and RMSE values for train and validation datasets.

| | | | **R** | **RMSE** | **MAPE** |
|---|---|---|---|---|---|
| $TSS_{Treated}$ | GSGMDH | Train (122 *) | 0.86 | 442 | 985 |
| | | Test (46) | 0.82 | 501 | 1131 |
| | MLR | Train (122) | 0.30 | 736 | 65 |
| | | Test (46) | 0.54 | 706 | 364 |
| $BOD_{Treated}$ | GSGMDH | Train (120) | 0.99 | 36 | 106 |
| | | Test (52) | 0.99 | 31 | 89 |
| | MLR | Train (120) | 0.83 | 159 | 558 |
| | | Test (52) | 0.67 | 199 | 443 |
| $COD_{Treated}$ | GSGMDH | Train (123) | 0.85 | 573 | 261 |
| | | Test (54) | 0.91 | 619 | 192 |
| | MLR | Train (123) | 0.54 | 890 | 820 |
| | | Test (54) | 0.73 | 575 | 943 |
| $TP_{Treated}$ | GSGMDH | Train (62) | 0.73 | 8.99 | 428 |
| | | Test (28) | 0.91 | 5.66 | 457 |
| | MLR | Train (62) | 0.59 | 6.40 | 58 |
| | | Test (28) | 0.80 | 12.8 | 63 |
| $TN_{Treated}$ | GSGMDH | Train (57) | 0.96 | 3.49 | 310 |
| | | Test (29) | 0.96 | 4.08 | 127 |
| | MLR | Train (57) | 0.58 | 7.60 | 68 |
| | | Test (29) | 0.70 | 10.9 | 54 |

* Number of samples used for training and testing the models.

Looking at Table 4, it is evident that the ranges of water quality parameters are highly variable as many different types of wash-waters were studied. The maximum value for $TSS_{Raw}$ stands out as it is very high, even compared to the maximum $TS_{Raw}$ value. This is because the $TS_{Raw}$ value corresponding to this high $TSS_{Raw}$ value was not available for some samples, as was predicted by Mundi et al. [25]. Some of the past data from other datasets only measured limited or different water quality parameters, and this was a major challenge, as some data points had to be excluded for some models. As a result, some models have lower sample numbers compared to others, as seen in Table 5.

The key findings of the study, that is the ability to predict treated water quality based on treatment, process, and raw wash-water quality parameters, are highlighted in Table 5. The table shows a number of samples used to construct the models, where training was 70% while test/validation was 30% of the total samples, and the selected input parameters.

Along with quality and performance parameters of the two different modeling methods, such as the coefficient of determination (R—%), the residual mean standard error (RMSE—mg/L), the and mean absolute percent error (MAPE—%).

A total of six water quality parameters were analyzed for MLR and GSGMDH as shown in Table 5. The greatest improvement was made for the $TSS_{Treated}$ model, where GSGMDH significantly improved the prediction, from $R^2$ of 30% in MLR to 90%, while $BOD_{Treated}$ was only improved by 16% using the GSGMDH method. The $R^2$ for TSS models were much lower when compared to Mjalli et al. [29] who reported values as high as 85%. The $R^2$ and RMSE improved for all six water quality parameters, while the MAPE did not show a consistent trend when going from MLR to the GSGMD modeling method. The use of both MAPE and RMSE was useful in providing a better understanding of the accuracy

of predicted values and sensitivity of outliers, which play an important role in over or underfitting models.

Overfitting happens when a model learns the data details and noise in the training data that negatively impacts the performance and the models' ability to generalize. Overfitting usually occurs with nonparametric and nonlinear models such as GSGMDH. In this case, we used a combination of both data details and noise, as some treatments had few experiments or single samples for full scale treatment, and due to the high variability of the data given different wash-water types and processes. Underfitting can also occur when the data does not show strong correlations or does not show a strong fit to linear functions.

Some underfitting is evident in MLR models as seen for $TSS_{Treated}$, where the $R^2$ is 30%, very low. The RMSE and MAPE show the magnitude and percentage of accuracy for each type of model. For the $TSS_{Treated}$ model, the RMSE and MAPE were higher and lower, respectively, when compared to GSGMDH. A reason for this was that the model was based on highly variable datasets, and as such, the accuracy of the MRL $TSS_{Treated}$ model is quite low. While with the use of GSGMDH the prediction was improved significantly, $R^2$ to 86%, the MAPE increased. The increase in the MAPE is attributed to the predicted values that are extreme, with large deviation from the actual value.

$TP_{Treated}$ and $TN_{Treated}$ models also improved as indicated by the change in $R^2$ from MLR to GSGMDH. $TP_{Treated}$ and $TN_{Treated}$ models were improved by approximately 14% and 38% points with respect to the $R^2$. The RMSE for all models for $TP_{Treated}$ and $TN_{Treated}$ was in the range of 3.49 to 8.99 mg/L, which is reasonably good, given the variability in wash-water data points. The MAPE ranges 54–457% for all models (Train and Test) of $TP_{Treated}$ and $TN_{Treated}$, reflecting the low quality of model accuracy, which is attributed to the low number of samples and the highly variable dataset. However, it is interesting to note that the MLR models had a lower MAPE compared to the GSGMDH. This is attributed to the fact that linear models can better absorb outlier and extreme predicted values.

The $BOD_{Treated}$ and $COD_{Treated}$ models showed the best fit and improvement with the use of GSGMDH methods, with $R^2$ increasing by 16% and 31%, respectively. In addition to showing improvements in RMSE and MAPE. The $BOD_{Treated}$ model improved by 16% as indicated by R. Measured and predicted values for treated BOD between MLR and GSGMDH models are show in Figure 2. This figure shows that MLR predicted values are widely scattered, while the GSGMDH model's predicted values are much closer to the actual or measured values. This is due to the higher number of input parameters used for GSGMDH models in addition to its advanced capabilities in modelling non-linear functions. The overall trends show that GSGMDH methods can provide better prediction than MLR methods, as shown by the improved $R^2$ for some of the developed models.

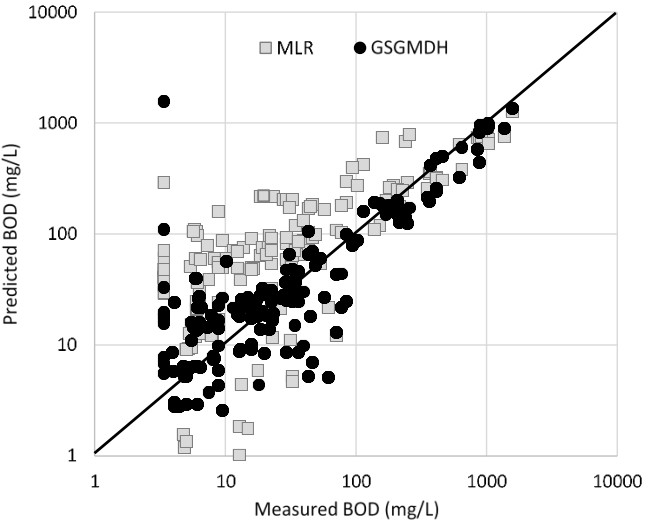

**Figure 2.** Measured versus Predicted treated BOD levels.

However, a fine balance is required between the number of input variables selected for models and the risk of overfitting. Given the challenges of the diversity and breadth of the samples and experiments studied, the models were still able to capture and show generalized solutions. This was due to the ability of the GSGMDH to capture nonlinear relationships better than MLR methods. However, the large RMSE and MAPE observed for some models can make it challenging to assess the predicted effluent water quality parameters against regulations for effluent release, which are typically very low, as noted in Table 4. In general, it is evident that both MLR and GSGMDH methods are feasible in determining the treated water quality of wash-water/wastewater from the fruit and vegetable industry, as highlighted here. MLR is a great modeling method as it provided insight into the impact of coefficients of the input parameters, while GSGMDH provides greater accuracy; however, it can exaggerate outliers and incorrectly predicted values. A high degree of variability exists, which is expected due to the non-linearity of the input variables. Another major challenge is that of large datasets, which are often required for GSGMDH models; however, this was not the case for this study, where many different wash-waters were studied under many different treatment scenarios. Modelling is best suited for a single treatment with a large dataset. This study attempted to formulate a universal model, and as observed in this paper, this can be very challenging due to the high level of error in some models, as shown by MAPE and RMSE. However, the models can be improved on by collecting additional data.

The MLR equations are presented in Table 2. All equations have a positive intercept followed by a negative treatment coefficient, followed by positive process coefficients for most models, while the rest of the coefficients show both positive and negative values. The process coefficients are negative for some models because their ranking for W and WP process is different for each model. The worst conditions were represented by the lower rank of 0.5 equivalent to the WP process, and for other water quality parameters, it was equivalent to W process. The parameters $TN_{Raw}$, $TDS_{Raw}$, and $NH4-N_{Raw}$ were most prevalent in the developed models. The magnitude and negative or positive value of the coefficient determine the level of effect on the predicted parameter. The GSGMDH equations are presented in Table 6. The structure of the GSMDH model for all studied parameters including TSS, BOD, COD, TP, and TN are shown in Figure 3a–e, respectively.

These MLR models/equations provide a simple, convenient, and practical way to determine the treatment effectiveness of the particular wash-waters, which can be utilized by producers/growers to assess for potential treatment to manage and reuse wash-waters. For example, $BOD_{Treated}$ can be determined by first selecting Equation (14) in Table 2, then obtaining the applicable treatment and process variables from Table 1, which will be multiplied by their corresponding coefficients. The $TDS_{Raw}$, $BOD_{Raw}$, and $TN_{Raw}$ are then plugged into the models for the wash-water/wastewater to be predicted. Once all the values are obtained, they are then entered into the $BOD_{Treated}$ model, as shown by Equation (14) found in Table 2. The calculated answer can be used along with corresponding RMSE and MAPE to assess treatment reduction/effectiveness.

To illustrate the use of the $BOD_{Treated}$ model, wash-water with TDS of 475 mg/L, $COD_{Raw}$ of 165 mg/L, a process type of W, and undergoing EC&F treatment would have $BOD_{Treated}$ of 60 mg/L. First, the numerical parameters are normalized; for TDSRaw, 475 is divided by the max value of $TDS_{Raw}$ (from Table 4) plus one, 475/(8740 + 1), which equals 0.0545. Similarly, $COD_{Raw}$ is normalized, and the value is 0.0133. The next step is to look up the process value for W type process from Table 1 for the $BOD_{Treated}$ model, which is 1, as well as for the EC&F treatment, which is 0.56. The next step is to substitute the values into Equation (14) from Table 2 to calculate $BOD_{Treated}$. In some cases, the calculated value could be negative, given the linear relationship, and should be converted to 1 for simplicity. The equations of MLR and GSGMDH can be summarized and integrated into an easy-to-use Microsoft excel worksheet tool. The user can input the raw wash-water values and obtain a comparative analysis of treated effluent for all treatments at once. The predicted values for each treatment are then compared with regulated effluent limits to understand which

treatment is most capable of meeting the standards. The worksheet also incorporates a cost analysis for the different treatments studied to provide cost/benefit analysis.

**Table 6.** The developed GSGMDH based equation for prediction.

| | **Input Parameters for all Following Models: x1 = Process, x2 = BOD$_{Raw}$, x3 = COD$_{Raw}$, x4 = NH$_4$−N$_{Raw}$, x5 = TN$_{Raw}$, x6 = TP$_{Raw}$, x7 = TSS$_{Raw}$, x8 = TS$_{Raw}$, x9 = TDS$_{Raw}$, x10 = Treatment** |
|---|---|
| TSS | TSS = (|1.528 − 30.083 ∗ x11 − 15.482 ∗ x1 − 4.522 ∗ x10 + 35.961 ∗ x1 ∗ x11 + 53.634 ∗ x10 ∗ x11 − 2.429 ∗ x10 ∗ x1 + 58.95 ∗ x11 ∗ x11 + 45.6 ∗ x1 ∗ x1 + 6.614 ∗ x10 ∗ x10 − 1.355 ∗ x10 ∗ x1 ∗ x11 + 5.404 ∗ x1 ∗ x11 ∗ x11 − 23.609 ∗ x1 ∗ x1 ∗ x11 − 107.196 ∗ x10 ∗ x11 ∗ x11 + 1.459 ∗ x10 ∗ x1 ∗ x1 − 33.867 ∗ x10 ∗ x10 ∗ x11 + 0.353 ∗ x10 ∗ x10 ∗ x1 − 49.415 ∗ x11 ∗ x11 ∗ x11 − 30.274 ∗ x1 ∗ x1 ∗ x1 − 2.809 ∗ x10 ∗ x10 ∗ x10|) ∗ 7160<br>Where: x11 = 0.137 − 0.639 ∗ x5 − 0.1419 ∗ x10 − 0.7245 ∗ x10 ∗ x5 + 3.46 ∗ x5 ∗ x5 − 0.0056 ∗ x10 ∗ x10 − 4.203 ∗ x10 ∗ x5 ∗ x5 + 1.914 ∗ x10 ∗ x10 ∗ x5 − 0.274 ∗ x5 ∗ x5 ∗ x5 − 0.0313 ∗ x10 ∗ x10 ∗ x10 |
| BOD | BOD = (|0.008 + 0.841 ∗ x21 + 0.044 ∗ x9 − 0.128 ∗ x5 − 3.982 ∗ x9 ∗ x21 − 0.17 ∗ x5 ∗ x21 − 1.682 ∗ x5 ∗ x9 + 5.71 ∗ x21 ∗ x21 + 0.0196 ∗ x9 ∗ x9 + 0.943 ∗ x5 ∗ x5 + 3.0926 ∗ x5 ∗ x9 ∗ x21 − 2.853 ∗ x9 ∗ x21 ∗ x21 + 5.668 ∗ x9 ∗ x9 ∗ x21 − 2.219 ∗ x5 ∗ x21 ∗ x21 − 5.524 ∗ x5 ∗ x9 ∗ x9 − 1.328 ∗ x5 ∗ x5 ∗ x21 + 11.933 ∗ x5 ∗ x5 ∗ x9 − 6.199 ∗ x21 ∗ x21 ∗ x21 + 0.218 ∗ x9 ∗ x9 ∗ x9 − 3.512 ∗ x5 ∗ x5 ∗ x5|) ∗ 2298 + 2<br>Where: x11 = −0.554 − 0.426 ∗ x10 + 0.42 ∗ x2 + 1.623 ∗ x1 − 1.652 ∗ x2 ∗ x10 + 0.224 ∗ x1 ∗ x10 + 1.517 ∗ x1 ∗ x2 + 0.724 ∗ x10 ∗ x10 + 1.265 ∗ x2 ∗ x2 − 0.766 ∗ x1 ∗ x1 + 1.322 ∗ x1 ∗ x2 ∗ x10 − 0.289 ∗ x2 ∗ x10 ∗ x10 + 1.185 ∗ x2 ∗ x2 ∗ x10 − 0.586 ∗ x1 ∗ x10 ∗ x10 − 1.6 ∗ x1 ∗ x2 ∗ x2 + 0.17 ∗ x1 ∗ x1 ∗ x10 − 1.447 ∗ x1 ∗ x1 ∗ x2 − 0.115 ∗ x10 ∗ x10 ∗ x10 − 0.622 ∗ x2 ∗ x2 ∗ x2 − 0.3 ∗ x1 ∗ x1 ∗ x1<br>and: x21 = −0.005 + 0.172 ∗ x11 + 0.071 ∗ x9 + 0.794 ∗ x2 + 7.808 ∗ x9 ∗ x11 + 25.691 ∗ x2 ∗ x11 − 3.26 ∗ x2 ∗ x9 − 21.654 ∗ x11 ∗ x11 − 0.7863662692 ∗ x9 ∗ x9 − 9.392 ∗ x2 ∗ x2 − 48.61 ∗ x2 ∗ x9 ∗ x11 + 17.523 ∗ x9 ∗ x11 ∗ x11 + 2.059 ∗ x9 ∗ x9 ∗ x11 − 17.201 ∗ x2 ∗ x11 ∗ x11 − 6.373 ∗ x2 ∗ x9 ∗ x9 − 20.94 ∗ x2 ∗ x2 ∗ x11 + 28.041 ∗ x2 ∗ x2 ∗ x9 + 43.899 ∗ x11 ∗ x11 ∗ x11 + 1.773 ∗ x9 ∗ x9 ∗ x9 + 8.693 ∗ x2 ∗ x2 ∗ x2 |
| COD | COD = = (|0.000423 + 0.073 ∗ x12 + 0.79 ∗ x11 − 6.087 ∗ x11 ∗ x12 + 1.742 ∗ x12 ∗ x12 + 5.405 ∗ x11 ∗ x11 − 9.309 ∗ x11 ∗ x12 ∗ x12 + 30.949 ∗ x11 ∗ x11 ∗ x12 − 0.46 ∗ x12 ∗ x12 ∗ x12 − 22.407 ∗ x11 ∗ x11 ∗ x11|) ∗ 8298 + 2<br>Where: x11 = −0.059 + 0.131 ∗ x10 + 1.3484 ∗ x9 − 0.694 ∗ x2 − 2.875 ∗ x9 ∗ x10 + 2.066 ∗ x2 ∗ x10 + 5.147 ∗ x2 ∗ x9 + 0.067 ∗ x10 ∗ x10 − 3.47 ∗ x9 ∗ x9 + 1.432 ∗ x2 ∗ x2 − 0.608 ∗ x2 ∗ x9 ∗ x10 + 1.315 ∗ x9 ∗ x10 ∗ x10 + 1.326 ∗ x9 ∗ x9 ∗ x10 − 1.818 ∗ x2 ∗ x10 ∗ x10 − 15.481 ∗ x2 ∗ x9 ∗ x9 − 0.94 ∗ x2 ∗ x2 ∗ x10 + 8.23 ∗ x2 ∗ x2 ∗ x9 − 0.104 ∗ x10 ∗ x10 ∗ x10 + 6.812 ∗ x9 ∗ x9 ∗ x9 − 3.527 ∗ x2 ∗ x2 ∗ x2<br>and: x12 = −1.119 − 6.106 ∗ x10 − 34.025 ∗ x2 − 5.377 ∗ x1 − 1.72 ∗ x2 ∗ x10 + 13 ∗ x1 ∗ x10 + 103.148 ∗ x1 ∗ x2 + 2.854 ∗ x10 ∗ x10 + 1.817 ∗ x2 ∗ x2 + 27.183 ∗ x1 ∗ x1 + 2.378 ∗ x1 ∗ x2 ∗ x10 − 0.7980132594 ∗ x2 ∗ x10 ∗ x10 + 0.834 ∗ x2 ∗ x2 ∗ x10 − 1.312 ∗ x1 ∗ x10 ∗ x10 − 1.213 ∗ x1 ∗ x2 ∗ x2 − 7.666 ∗ x1 ∗ x1 ∗ x10 − 69.104 ∗ x1 ∗ x1 ∗ x2 − 0.924 ∗ x10 ∗ x10 ∗ x10 − 1.167 ∗ x2 ∗ x2 ∗ x2 − 20.56 ∗ x1 ∗ x1 ∗ x1 |
| TP | TP = (| −0.055 − 1.609 ∗ x11 + 0.619 ∗ x10 + 1.528 ∗ x4 − 10.509 ∗ x10 ∗ x11 + 26.087 ∗ x4 ∗ x11 − 1.137 ∗ x4 ∗ x10 + 51.713 ∗ x11 ∗ x11 − 0.738 ∗ x10 ∗ x10 − 16.005 ∗ x4 ∗ x4 + 37.421 ∗ x4 ∗ x10 ∗ x11 − 42.84 ∗ x10 ∗ x11 ∗ x11 + 8.358 ∗ x10 ∗ x10 ∗ x11 − 929.104 ∗ x4 ∗ x11 ∗ x11 − 0.0658 ∗ x4 ∗ x10 ∗ x10 + 473.109 ∗ x4 ∗ x4 ∗ x11 − 3.534 ∗ x4 ∗ x4 ∗ x10 + 322.282 ∗ x11 ∗ x11 ∗ x11 + 0.263 ∗ x10 ∗ x10 ∗ x10 − 39.329 ∗ x4 ∗ x4 ∗ x4|) ∗ 89.96 + 0.04<br>where: x11 = −2.442 + 7.202 ∗ x1 + 0.533 ∗ x4 + 0.896 ∗ x4 ∗ x1 − 4.397 ∗ x1 ∗ x1 − 3.009 ∗ x4 ∗ x4 − 4.066 ∗ x4 ∗ x1 ∗ x1 + 21.622 ∗ x4 ∗ x4 ∗ x1 − 0.2734052233 ∗ x1 ∗ x1 ∗ x1 − 15.831 ∗ x4 ∗ x4 ∗ x4 |
| TN | TN = (|0.079 − 0.157 ∗ x11 + 0.747 ∗ x5 − 1.742 ∗ x9 + 8.214 ∗ x5 ∗ x11 − 5.488 ∗ x9 ∗ x11 − 10.891 ∗ x9 ∗ x5 + 1.884 ∗ x11 ∗ x11 + 2.472 ∗ x5 ∗ x5 + 14.098 ∗ x9 ∗ x9 + 3.182 ∗ x9 ∗ x5 ∗ x11 − 4.938 ∗ x5 ∗ x11 ∗ x11 − 6.586 ∗ x5 ∗ x5 ∗ x11 − 8.161 ∗ x9 ∗ x11 ∗ x11 − 19.931 ∗ x9 ∗ x5 ∗ x5 + 7.19 ∗ x9 ∗ x9 ∗ x11 + 45.043 ∗ x9 ∗ x9 ∗ x5 + 1.522 ∗ x11 ∗ x11 ∗ x11 + 0.609 ∗ x5 ∗ x5 ∗ x5 − 22.436 ∗ x9 ∗ x9 ∗ x9|) ∗ 60.83 + 0.03<br>Where: x11 = 0.124 + 0.296 ∗ x4 + 0.339 ∗ x9 − 0.06 ∗ x10 − 7.27 ∗ x9 ∗ x4 − 7.826 ∗ x10 ∗ x4 + 0.627 ∗ x10 ∗ x9 + 42.361 ∗ x4 ∗ x4 − 1.624 ∗ x9 ∗ x9 − 0.143 ∗ x10 ∗ x10 + 18.159 ∗ x10 ∗ x9 ∗ x4 − 196.991 ∗ x9 ∗ x4 ∗ x4 + 71.295 ∗ x9 ∗ x9 ∗ x4 + 0.209 ∗ x10 ∗ x4 ∗ x4 + 0.382 ∗ x10 ∗ x9 ∗ x9 + 1.765 ∗ x10 ∗ x10 ∗ x4 − 1.498 ∗ x10 ∗ x10 ∗ x9 − 2.397 ∗ x4 ∗ x4 ∗ x4 + 0.785 ∗ x9 ∗ x9 ∗ x9 + 0.258 ∗ x10 ∗ x10 ∗ x10 |

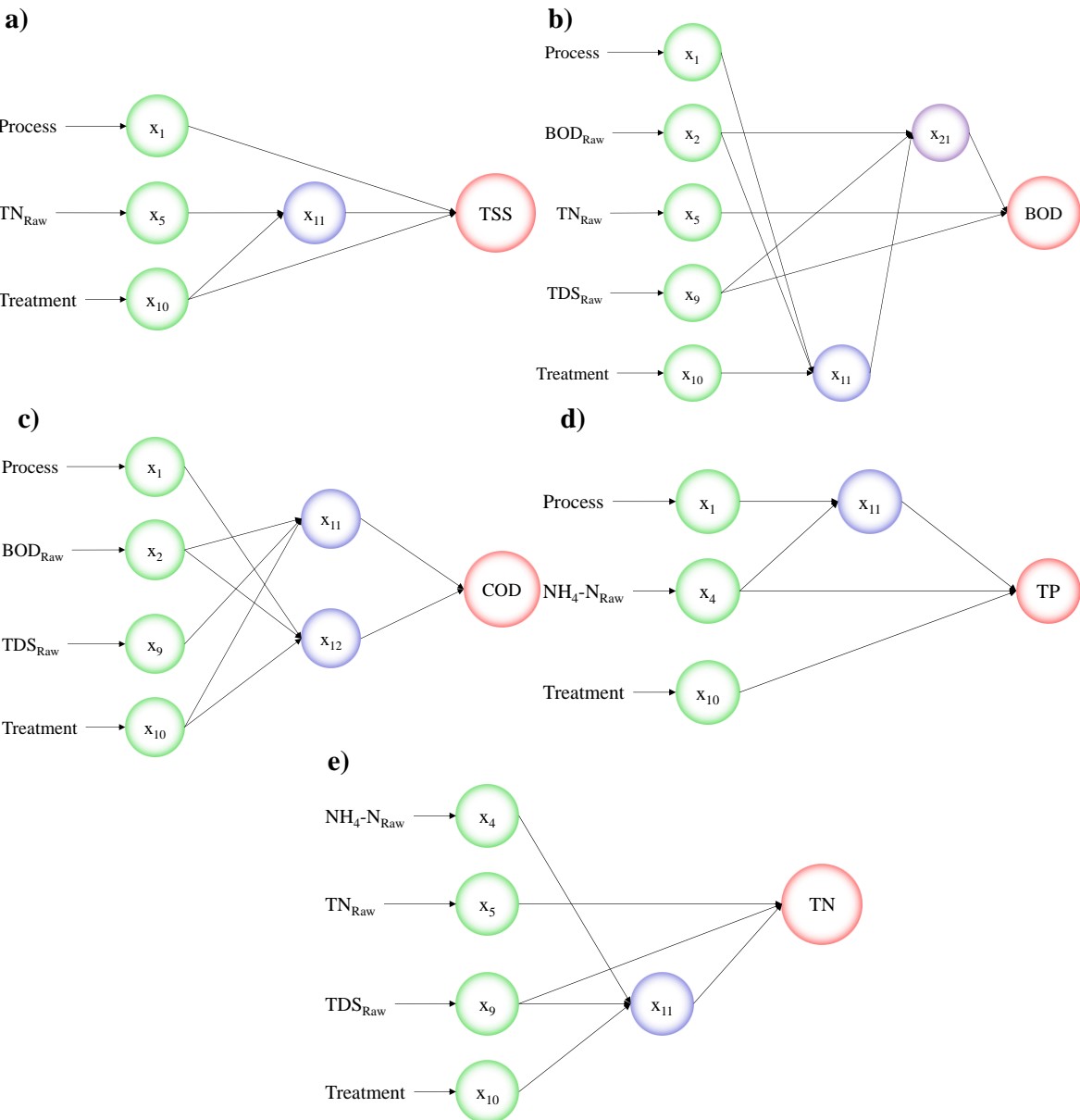

**Figure 3.** Structure of the developed GSGMDH models: (**a**) TSS; (**b**) BOD; (**c**) COD); (**d**) TP; and (**e**) TN.

The developed models, which complement the treatment decision matrices/tables produced in Mundi et al. [24], show the level of treatment expected from various wash-waters/wastewaters. The decision matrices/tables provided a range for treatment effectiveness, while these models extend that analysis to numerical models for more flexible treatment predictions. Some limitation exists, such as over and underestimating treated water quality parameters due to under and overfitting of the models. However, the prediction of treated water quality levels is still valuable, and the methods demonstrated herein can be implemented by facilities for continuous monitoring and treatment selections. Many predicative tools utilize the black box ANN models, which are cumbersome to utilize that require software and cannot easily produce equations to use, similar to the MLR and GSGMDH shown in this study. This information is valuable for farmers, governments, engineers, and consultants, and other stakeholders in determining wash-water treatment and sizing, understanding treatments capable of meeting regulatory standards, and treatment costs from predicted water quality parameters. The study was successful in capturing a

wide variety of information and concatenating it into a useful tool for making an important decision on the treatment of wash-waters/wastewaters, which previously did not exist.

## 4. Conclusions

The study highlighted the use of MLR and GSGMDH techniques to model fruit and vegetable washing and processing wastewater treatments in predicting treated water quality parameters. These universal models successfully captured data from many different commodities representatives of post-harvest processing, including the fresh-cut industry. Modelling a wastewater treatment process is difficult to accomplish due to the high non-linearity of the treatments studied and the non-uniformity and variability of wash-water/wastewaters as well as the nature of the chemical/biological reactions occurring in treatments. An MLR model approach was assessed by the authors of this study to understand any linear relationships, and the GSGMDH models for understanding interdependency between outputs and inputs involving nonlinear relationships. The MLR model utilizes simple linear equations to describe relationships, which can be very useful in predicting treatment levels as shown above. Similarly, GSGMDH models are non-linear, hence the more complicated equations when compared to the linear models but provide a more accurate prediction. For example, different stakeholders (design engineers, operators, suppliers, government, local fruit and vegetable organization/clubs) can utilize the above-developed models to determine which physio-chemical treatment can be utilized to treat various wash-water/wastewater types, and its corresponding approximate effluent expected water quality levels, without having to collect any data, spend money on testing water samples, or perform intensive studies before considering implementation of the technology.

The models developed for estimating treated parameters from bench-scale and full-scale treatments include TSSTreated, $BOD_{Treated}$, $COD_{Treated}$, $TN_{Treated}$, $TP_{Treated}$, and $NH4-N_{Treated}$ water quality parameters. The derived models performed very well as indicated by $R^2$ values. The RMSE performance parameter indicates the expected error boundaries, which is in the range of acceptable. The variability is inherent to the developed models, as the datasets were a combination of lab and industry-wide samples and can be improved by increasing sample size for various sets, such as for full-scale treatments. A combination of data types, number of samples, the variability of the different wash-water types, and noise within the data influence the quality of the models. The balance between a number of input variables selected for modelling and risk of over or underfitting is critical.

Previous research has not consider the holistic approach shown in this study for understanding different fruit and vegetable wastewater treatments. As such, for the first time this study considered 14 different facilities (Apple, Carrot, Potato, Ginseng, and others) and 13 different wash-waters/wastewaters treatments for treatment effectiveness of TSS, TN, TP, NH4-N, COD, and BOD water quality parameters. This in itself is very novel, as the study has successfully researched and developed a complex problem into a workable solution, combining multiple factors while maintaining the depth and breadth of the study topic. MLR models were a great tool for predictions as they provided insight into the impact of input parameters through their coefficients and showed consistent error over all treatments. GSGMDH was more flexible and better captured the results but required greater caution when predicting values. Another novelty in this study is that of providing tools to estimate the treatment and corresponding water quality of major water quality parameters (COD, BOD, TSS, etc.). No tool existed in the literature of the fruit and vegetable washing and processing sectors that can easily predict water quality of treated waters. These novel MLR and GSGMDH models for predicting wash-water treatment feasibility provide a long list of treatments, which was not previously available or studied.

**Author Contributions:** Conceptualization, G.M., R.G.Z., K.W., H.B. and B.G. Investigation and Methodology, G.M., H.B. and B.G.; Formal Analysis and Validation, G.M., H.B. and B.G. Visualization, G.M., R.G.Z., K.W. and H.B.; Project administration and funding acquisition, R.G.Z.; Writing, review

and editing, G.M., R.G.Z., K.W., H.B. and B.G. All authors have read and agreed to the published version of the manuscript.

**Funding:** This research was funded by Ontario Ministry of Agriculture and Rural Affairs (OMAFRA), grant number 052111.

**Institutional Review Board Statement:** Not applicable.

**Informed Consent Statement:** Not applicable.

**Data Availability Statement:** Data used in developing the various models comes from a variety of sources and due to confidentially by some producers is not available because of competitive interests.

**Conflicts of Interest:** The authors declare no conflict of interest.

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
