# Peer review of "Machine Learning Models for Predicting Water Quality of Treated Fruit and Vegetable Wastewater"

_water, doi:10.3390/w13182485_

Round 1
Reviewer 1 Report
The novelty of the research is well described, but still poor in practical examples of application. The main concern deals with the presentation of materials and methods and, consequently, of results. More figures are needed and with higher quality. An overall warning: PLEASE PAY ATTENTION TO THE WATE MDPI TEMPLATE.
In detail, you have here my comments:
ABSTRACT: Please correct R2 in R2;
KEYWORDS: Please replace the key-sentences in keywords;
INTRODUCTION: Some more practical examples are needed to explain the complete scientific background. For example, refers to water quality control also in reclamation channels (i.e., Lama, G.F.C.; Crimaldi, M.; Pasquino, V.; Padulano, R.; Chirico, G.B. Bulk Drag Predictions of Riparian Arundo donax Stands through UAV-acquired Multispectral Images. Water 2021, 13, 1333. https://doi.org/10.3390/w13101333);
MATERIALS AND METHODS
Please add to each sub-section a figure to explain each method, and pay attention to the template for equations.
RESULTS AND DISCUSSION
Please add more figures to explain your very interesting findings and pay attention to the template for equations. Table 6 is too big, the reader is not supported at all in understanding all the long and tedious equations and symbols.
For these reasons, I suggest reconsidering the article after major revision
Author Response
Reviewer # 1:
The novelty of the research is well described, but still poor in practical examples of application. The main concern deals with the presentation of materials and methods and, consequently, of results. More figures are needed and with higher quality. An overall warning: PLEASE PAY ATTENTION TO THE WATE MDPI TEMPLATE.
Response: Thank you. We are very thankful for your kind and constructive suggestions and comments. We also appreciate the time and effort that you have dedicated to provide valuable feedback on our work. We carefully considered the comments. The corrections are highlighted by track changed, and the detailed response to comments is addressed below.
In detail, you have here my comments:
ABSTRACT: Please correct R2 in R2;
Response: Thank you. The requested revision was accordingly done. (Please see Abstract)
KEYWORDS: Please replace the key-sentences in keywords;
Response: Thank you. The requested revision was accordingly done. (Please see Keywords)
INTRODUCTION: Some more practical examples are needed to explain the complete scientific background. For example, refers to water quality control also in reclamation channels (i.e., Lama, G.F.C.; Crimaldi, M.; Pasquino, V.; Padulano, R.; Chirico, G.B. Bulk Drag Predictions of Riparian Arundo donax Stands through UAV-acquired Multispectral Images. Water 2021, 13, 1333.https://doi.org/10.3390/w13101333);
Response: Thank you. The requested revision was accordingly done. (Please see Introduction)
MATERIALS AND METHODS
Please add to each sub-section a figure to explain each method, and pay attention to the template for equations.
Response: Thank you. The requested revision was accordingly done. (Please see Equations)
RESULTS AND DISCUSSION
Please add more figures to explain your very interesting findings and pay attention to the template for equations. Table 6 is too big, the reader is not supported at all in understanding all the long and tedious equations and symbols.
Response: Thank you. Thank you for the comment. Table 6 was revised and simplified. In this Table an explicit equation for predicting treatment effluent water quality of wash-water treatments. These equations can be applied to the practical cases. In order to make this Table more clear, new Figure that shows the structure of the developed models was added in the manuscript. (Please see Table 6 & Figure 3)
For these reasons, I suggest reconsidering the article after major revision

Reviewer 2 Report
The manuscript “Machine Learning Models for Predicting Water Quality of Treated Fruit and Vegetable Wastewater” presents two different types of models that can be used to predict the treatment effectiveness of different methods based on a number of input parameters. The paper presents both a standard MLR approach a non-linear GMDH method for modeling that provides model interpretability. This two different types of models are not referenced in the abstract.
General Notes:
There is a large number of issues with capitalization in the paper – many are highlighted in the attached markup. There are also a number of typos and phrases that seem to be missing a word, e.g., L41 and L74 (highlighted in the markup). I encourage the authors to read the manuscript carefully to address these issues. I did not mark all instances in my review
The acronyms are defined multiple times, e.g., L91-92.
The paper uses passive voice in some places where it makes the meaning difficult to understand. For example, L259 says that the MATLAB model was coded. Did the authors code the model or was the model coded by someone else and provided on-line or elsewhere. First person is much clearer on these issues, i.e., “We coded the ….”.
Technical Issues
Variable normalization, the paper uses X_i/(X_max + 1) and states this gives a range of 0-1. It does not, for example if the data go from 1 to 3, the largest normalized value would be 0.75. Why is this normalization used? Should cite papers if using non-standard methods. The standard method is:
(X_i – X_min)/(X_max – X_min)
Significant figures. The equations in tables 5 and 6 are almost unreadable because of significant figures (9 places after the decimal point). If the models require this level of precision in the parameters, that is an issue.
Table 6 would be easier to read if the equations were given as a general equation with a table of coefficients. If this could be done it would be easier to read and apply the models
The paper implies there is a spreadsheet version of this model. The paper does not mention where or if this model is available.
L489 States that values of treated BOD that are negative be changed to 1. Why 1 and not 0 or some small number. Also, the graph (Figure 2) seems to have a large number of measured BOD values with the same value, I would guess about 7, though it is difficult to estimate in a log-log plot. Why are all these minimum values the same and well above quantitation limits for most tests?
L53-55 Not every problem is caused by climate change, especially this one. It seems every author tries to put this statement in their papers. I suggest deleting – if not, provide a citation or data showing how this issue is driven by climate change.
Specific comments:
L41 English construction
L71 Capitalization
L74 should “serving” be “several”?
L77 capitalization
L91-92 Acronyms previously defined
L135-136 capitalization
L171 “data was” should be “data were”, data is plural, datum is singular
L173 capitalization R software is always capitalized
L180 capitalization – when it is someones name
L180 should “person’s” be “pearson’s”
L191 Eq 1 discussed above. This is non-standard and does provide a 0-1 range.
L204 capitalization
L210 acronym previously defined
L259 who coded the model (discussed above)
L270 missing number for the 3rd item in the list
L281 Figure 1 “Keep neurons with higher RMSE than criteria”, should this be lower?
L296-~305 or so. This has been stated already nearly verbatim.
L331 Table 1 How did you compute normalized values of 1 using Equation 1 – it isn’t possible
L351 capitalizatnoi
L351 should “person’s” be “pearson’s”?
L356 states that variables were selected. Who selected them? If it was the authors, this should be stated, e.g., “We selected…”
L432 Figure 2 Why all the measured values at about 7?
L432 Figure 2, based on line 489, is the single predicted value of 1 actually a negative value?
L468-470 Significant figures
L492 States that this can be implemented in a spreadsheet. In the next lines it discussed how a user would use the spreadsheet. Was a spreadsheet created? Is the spreadsheet available? This should be discussed in the manuscript.
L514 delete the word “novel”
L521 “approach was assessed…” by who, someone in the literature, then cite it, the authors? Then state it.
L537 This list of specific fruits and vegetables appears here for the first time, it should be also stated in the section discussing the data.
L542 states that the GSGMDH model requires “greater caution”. Why does it require greater caution? What should users be cautious about? This was not discussed, only brought up here.

Author Response
Reviewer # 2:
The manuscript “Machine Learning Models for Predicting Water Quality of Treated Fruit and Vegetable Wastewater” presents two different types of models that can be used to predict the treatment effectiveness of different methods based on a number of input parameters. The paper presents both a standard MLR approach a non-linear GMDH method for modeling that provides model interpretability.
Response: Thank you. We are very thankful for your kind and constructive suggestions and comments. We also appreciate the time and effort that you have dedicated to provide valuable feedback on our work. We carefully considered the comments. The corrections are highlighted by track changed, and the detailed response to comments is addressed below.
This two different types of models are not referenced in the abstract.
Response: Thank you. The requested revision was accordingly done. (Please see Abstract)
General Notes:
There is a large number of issues with capitalization in the paper – many are highlighted in the attached markup. There are also a number of typos and phrases that seem to be missing a word, e.g., L41 and L74 (highlighted in the markup). I encourage the authors to read the manuscript carefully to address these issues. I did not mark all instances in my review
Response: Thank you. The requested revision was accordingly done. Please see track changes.
The acronyms are defined multiple times, e.g., L91-92.
Response: Thank you. The requested revision was accordingly done. Please see track changes.
The paper uses passive voice in some places where it makes the meaning difficult to understand. For example, L259 says that the MATLAB model was coded. Did the authors code the model or was the model coded by someone else and provided on-line or elsewhere. First person is much clearer on these issues, i.e., “We coded the ….”.
Response: Thank you. The requested revision was accordingly done. Please see track changes.
Technical Issues
Variable normalization, the paper uses X_i/(X_max + 1) and states this gives a range of 0-1. It does not, for example if the data go from 1 to 3, the largest normalized value would be 0.75. Why is this normalization used? Should cite papers if using non-standard methods. The standard method is:
(X_i – X_min)/(X_max – X_min)
Response: Thank you. The requested revision was accordingly done. The paper has been cited. The formula used provide a more robust scaling when considering the large difference in magnitude for the different water quality parameters. The formula does produce a number between 0-1, for example when looking at Raw value of ammonia, the min. is 0.09 mg/l and the max. value is 35 mg/L, so for a value of 35 mg/L the normalized value would be =35/(35+1) = 0.97, however for raw total solids of 13,805 mg/l the normalized value would be =13,805(13805+1)=0.9999.
Significant figures. The equations in tables 5 and 6 are almost unreadable because of significant figures (9 places after the decimal point). If the models require this level of precision in the parameters, that is an issue.
Response: Thank you. The requested revision was accordingly done. Please see track changes.
Table 6 would be easier to read if the equations were given as a general equation with a table of coefficients. If this could be done it would be easier to read and apply the models
Response: Thank you. The requested revision was accordingly done. Please see track changes.
The paper implies there is a spreadsheet version of this model. The paper does not mention where or if this model is available.
Response: Thank you. No spreadsheet has been developed. Just a recommendation for stakeholders.
L489 States that values of treated BOD that are negative be changed to 1. Why 1 and not 0 or some small number. Also, the graph (Figure 2) seems to have a large number of measured BOD values with the same value, I would guess about 7, though it is difficult to estimate in a log-log plot. Why are all these minimum values the same and well above quantitation limits for most tests?
Response: Thank you. The use of log scale to represent the values mean that a zero value would not easily be illustrated in Figure 2, as such recommendation was made to change values from 0 to 1. Since linear models are based on a linear equation, there may be situation where the models may produce a negative value result, so to correct this anomaly values may require the addition of standard deviation to bring the value into a range as such convert it to a positive value. The values in Figure 2 that are measured to be around 7 mg/L, but show very different predicted values are a results of wash-water/wastewater, which have a very high oxygen demand due to the intensive recirculation of wash-water for washing fruits such as apple, that are low in other water quality parameters such as solids, and do not match up with the rest of the generalized data set (wash-waters/wastewater), and therefore not easily captured by the model.
L53-55 Not every problem is caused by climate change, especially this one. It seems every author tries to put this statement in their papers. I suggest deleting – if not, provide a citation or data showing how this issue is driven by climate change.
Specific comments:
L41 English construction
L71 Capitalization
L74 should “serving” be “several”?
L77 capitalization
L91-92 Acronyms previously defined
L135-136 capitalization
L171 “data was” should be “data were”, data is plural, datum is singular
L173 capitalization R software is always capitalized
L180 capitalization – when it is someones name
L180 should “person’s” be “pearson’s”
L191 Eq 1 discussed above. This is non-standard and does provide a 0-1 range.
L204 capitalization
L210 acronym previously defined
L259 who coded the model (discussed above)
L270 missing number for the 3rd item in the list
L281 Figure 1 “Keep neurons with higher RMSE than criteria”, should this be lower?
L296-~305 or so. This has been stated already nearly verbatim.
L331 Table 1 How did you compute normalized values of 1 using Equation 1 – it isn’t possible
L351 capitalizatnoi
L351 should “person’s” be “pearson’s”?
L356 states that variables were selected. Who selected them? If it was the authors, this should be stated, e.g., “We selected…”
L432 Figure 2 Why all the measured values at about 7?
L432 Figure 2, based on line 489, is the single predicted value of 1 actually a negative value?
L468-470 Significant figures
L492 States that this can be implemented in a spreadsheet. In the next lines it discussed how a user would use the spreadsheet. Was a spreadsheet created? Is the spreadsheet available? This should be discussed in the manuscript.
L514 delete the word “novel”
L521 “approach was assessed…” by who, someone in the literature, then cite it, the authors? Then state it.
L537 This list of specific fruits and vegetables appears here for the first time, it should be also stated in the section discussing the data.
L542 states that the GSGMDH model requires “greater caution”. Why does it require greater caution? What should users be cautious about? This was not discussed, only brought up here.
Response: Thank you for the detailed feedback as noted above. The requested revision was accordingly done. Please see track changes.

Reviewer 3 Report
Machine Learning Models for Predicting Water Quality of Treated Fruit and Vegetable Wastewater
Abstract should show the main novel of this research
The different references should be discussed in the manuscript not only cited. (e.g., line 44, line 46, line 72, line 78) Review all manuscript.
Different optimization methods are defined MLR, ANN, among others. However, in my opinion this background section is significant for the research. I recommend to authors using the following references (and they can search more references) to complete the different optimization methods in the improvement of the quality and sustainability in wastewater.
Optimization strategies for the design and synthesis of distributed wastewater treatment networks. Computers & Chemical Engineering, 23, S161-S164.
On the use of linear models for the design of water utilization systems in process plants with a single contaminant. Trans IChemE, 79(Part A), 600-610.
Analysis of a wastewater treatment plant using fuzzy goal programming as a management tool: A case study. Journal of Cleaner Production, 180, 20-33.
Sewage treatment process refinement and intensification using multi-criteria decision making approach: a case study J. Water Process Eng., 37 (2020), Article 101485
Optimization of water networks in industrial processes. Journal of Cleaner Production, 17(9), 857-862.
The introduction should show clearly the main goal of the research
Section 2.1. The authors could do a table where they show the different samples and its description. It helps readers to understand better the description between lines 133-168
Sections 2.2, 2.3, 2.4 and 2.5 should be joined. Besides, the authors should join the different subsections in a unique subsection in which the authors describe figure 1 ( I think it should be located on first and later, they explain the rest of the equations) The different steps of figure 1 should be enumerated and described in the document
I think the authors should increase the measure of the evaluation of the model (only R2, RMSE and MAPE, it is not enough).
Figure 2 can be improved to clarify the fat points
Table 6, simplify the equations (e.g., x1*x1*x1……..) and use formula expression to show it. Multiply the different numbers for example (2298*|0.00815470959); is it necessary the use 11 decimals? These formulas are low useful with this format.
The authors should develop a result section in which they develop a deep discussion of the results, not only put table and figures.
The conclusion should show the applicability of the model to real cases and the difference with others methods. Besides, the should show clearly the novel of the research
Author Response
Reviewer # 3:
Machine Learning Models for Predicting Water Quality of Treated Fruit and Vegetable Wastewater
Abstract should show the main novel of this research
Response: Thank you. The requested revision was accordingly done. (Please see Abstract)
The different references should be discussed in the manuscript not only cited. (e.g., line 44, line 46, line 72, line 78) Review all manuscript.
Response: Thank you. The requested revision was made. (Please see Introduction)
Different optimization methods are defined MLR, ANN, among others. However, in my opinion this background section is significant for the research. I recommend to authors using the following references (and they can search more references) to complete the different optimization methods in the improvement of the quality and sustainability in wastewater.
Optimization strategies for the design and synthesis of distributed wastewater treatment networks. Computers & Chemical Engineering, 23, S161-S164.
On the use of linear models for the design of water utilization systems in process plants with a single contaminant. TransI ChemE, 79(Part A), 600-610.
Analysis of a wastewater treatment plant using fuzzy goal programming as a management tool: A case study. Journal of Cleaner Production, 180, 20-33.
Sewage treatment process refinement and intensification using multi-criteria decision making approach: a case study J. Water Process Eng., 37 (2020), Article 101485
Optimization of water networks in industrial processes. Journal of Cleaner Production, 17(9), 857-862.
Response: Thank you. The requested revision about the literature review was done. (Please see Introduction)
The introduction should show clearly the main goal of the research
Response: Thank you. The requested revision was accordingly done. (Please see Introduction)
Section 2.1. The authors could do a table where they show the different samples and its description. It helps readers to understand better the description between lines 133-168
Response: Thank you. Table 1 highlights all these details. In materials and method, we present general information about the parameters and the raw wash-waters treatments techniques. (Please see Table 1)
Sections 2.2, 2.3, 2.4 and 2.5 should be joined. Besides, the authors should join the different subsections in a unique subsection in which the authors describe figure 1 ( I think it should be located on first and later, they explain the rest of the equations) The different steps of figure 1 should be enumerated and described in the document
Response: Thank you for the comment. Sections 2.3 and 2.4 were combined. The descript of the Figure 1 was added in manuscript. (Please see Section 2)
I think the authors should increase the measure of the evaluation of the model (only R2, RMSE and MAPE, it is not enough).
Response: Thank you. The requested revision was accordingly done. (Please see Table 4)
Figure 2 can be improved to clarify the fat points
Response: Thank you for the comment. The Figures 1 & 2 were modified. (Please see Figures 1 & 2)
Table 6, simplify the equations (e.g., x1*x1*x1……..) and use formula expression to show it. Multiply the different numbers for example (2298*|0.00815470959); is it necessary the use 11decimals? These formulas are low useful with this format.
Response: Thank you for the comment and it has been revised and simplified. In addition, in order to make this Table more clear, new Figure that shows the structure of the developed GSGMDH models was added in the manuscript. (Please see Table 6 & Figure 3)
The authors should develop a result section in which they develop a deep discussion of the results, not only put table and figures.
Response: Thank you for the comment and feedback. Revision have been made and results are now compared to the other similar studies. The authors believe that a deep discussion of the results is now presented, there is a very limited number of studies in literature that deal with wastewater from fruit and vegetable processing, as such it is difficult to compare results with other studies. As such, this research is highly valuable and when published would be of great value to the sector/field.
The conclusion should show the applicability of the model to real cases and the difference with others methods. Besides, the should show clearly the novel of the research
Response: Thank you for the comment and feedback. Revision have been made to the conclusion section. Novelty of the study has been reiterated.

Reviewer 4 Report
This study applied multiple linear regression and the generalized structure of the group method of data handling to predict water quality parameters of wash-waters and wastewaters. This manuscript is well written and for readers to understand easily. I recommend this study is qualified to be published after the revision.
Comments
-In line 91 remove “ group method of data handling”. It is already abbreviated.
-In Table 4, R should be replaced with “R2”.
-R2 or R2 should make a consistent notation in the entire manuscript.
Author Response
Reviewer # 4:
This study applied multiple linear regression and the generalized structure of the group method of data handling to predict water quality parameters of wash-waters and wastewaters. This manuscript is well written and for readers to understand easily. I recommend this study is qualified to be published after the revision.
Comments
-In line 91 remove “ group method of data handling”. It is already abbreviated.
Response: Thank you. The requested revision was accordingly done. (Please see Introduction)
-In Table 4, R should be replaced with “R”.
Response: Thank you. The requested revision was accordingly done. (Please see Table 4)
-R or R2 should make a consistent notation in the entire manuscript.
Response: Thank you. It was done. (Please see Manuscript)

Round 2
Reviewer 1 Report
The article was improved in a very satisfactory way. After the revisions, the quality of the scientific message is more clear and direct, and the overall paper is more attractive to the reader.
There are very few issues to be improved:
- Please pay attention to using R or R2 correctly in the text. The reader must be sure that all the symbols have only one meaning.
- INTRODUCTION: Please mention also the importance of Machine Learning in previous works on Water quality (Granata, F.; Papirio, S.; Esposito, G.; Gargano, R.; De Marinis, G. Machine Learning Algorithms for the Forecasting of Wastewater Quality Indicators. Water 2017, 9, 105. https://doi.org/10.3390/w902010);
- INTRODUCTION: Please mention also the importance of Machine Learning in the hydrodynamic modeling of vegetated waterways, in predictive flow resistance research (Lama, G.F.C.; Errico, A.; Francalanci, S.; Solari, L.; Preti, F.; Chirico, G.B. Evaluation of Flow Resistance Models Based on Field Experiments in a Partly Vegetated Reclamation Channel. Geosciences 2020, 10, 47. https://doi.org/10.3390/geosciences10020047)
I accept the article after minor revision.
Author Response
The article was improved in a very satisfactory way. After the revisions, the quality of the scientific message is more clear and direct, and the overall paper is more attractive to the reader.
There are very few issues to be improved:
Response: Thank you very much for your time in reviewing the paper and providing recommendations. We are very thankful for your kind consideration of our paper for your prestigious journal. The corrections are highlighted by track changes, and the detailed response to comments is addressed below.
Please pay attention to using R or R2 correctly in the text. The reader must be sure that all the symbols have only one meaning.
Response: Thank you. The requested revision was accordingly done.
INTRODUCTION: Please mention also the importance of Machine Learning in previous works on Water quality (Granata, F.; Papirio, S.; Esposito, G.; Gargano, R.; De Marinis, G. Machine Learning Algorithms for the Forecasting of Wastewater Quality Indicators. Water 2017, 9, 105. https://doi.org/10.3390/w902010);
INTRODUCTION: Please mention also the importance of Machine Learning in the hydrodynamic modeling of vegetated waterways, in predictive flow resistance research (Lama, G.F.C.; Errico, A.; Francalanci, S.; Solari, L.; Preti, F.; Chirico, G.B. Evaluation of Flow Resistance Models Based on Field Experiments in a Partly Vegetated Reclamation Channel. Geosciences 2020, 10, 47. https://doi.org/10.3390/geosciences10020047)
Response: Thank you. The requested revision was accordingly done.
Reviewer 3 Report
The authors clarified the different suggestions.
Author Response
The authors clarified the different suggestions.
Response: Thank you very much for you time in reviewing the paper and providing recommendations. We are very thankful for your kind consideration of our paper for your prestigious journal.